# DNACHUNKER: LEARNABLE TOKENIZATION FOR DNA LANGUAGE MODELS

## ABSTRACT

DNA language models have emerged as powerful tools for decoding the complex language of DNA sequences. However, the performance of these models is heavily affected by their tokenization strategy, i.e., a method used to parse DNA sequences into a shorter sequence of chunks. In this work, we propose DNACHUNKER, which integrates a learnable dynamic DNA tokenization mechanism and is trained as a masked language model. Adopting the dynamic chunking procedure proposed by Hwang et al. (2025), our model learns to segment sequences into variable-length chunks. This dynamic chunking offers two key advantages: it's resilient to shifts and mutations in the DNA, and it allocates more detail to important functional areas. We demonstrate the performance of DNACHUNKER by training it on the human reference genome (HG38) and testing it on the Nucleotide Transformer and Genomic benchmarks. Further ablative experiments reveal that DNACHUNKER learns tokenization that grasps biological "grammar" and uses smaller chunks to preserve detail in important functional elements such as promoters and exons, while using larger chunks for repetitive, redundant regions.

## 1 INTRODUCTION

DNA sequences are the fundamental blueprint of life, containing the information that governs complex biological processes such as gene regulation (Moore et al., 2020), protein synthesis (Jia et al., 2024), DNA replication (Ekundayo & Bleichert, 2019), to name a few. Rapid advances in sequencing technology (Behjati & Tarpey, 2013) have made genomic data massively available. However, understanding and predicting the function encoded within these sequences remains a major challenge. The immense length and intricate nature of genomic data, along with a lack of high-quality, task-specific datasets, make it difficult to understand the underlying rules of this biological code.

Inspired by the success of large language models (LLMs; Anil et al., 2023), several recent works have begun investigating DNA language models (Ji et al., 2021; Sanabria et al., 2024; Dalla-Torre et al., 2025), moving beyond traditional rule-based methods to learn the "grammar" and "semantics" of DNA. In particular, the presence of long-range interactions between nucleotides and functional elements such as promoters and enhancers that act as "words" in the genomic language highlights the need for a tokenization strategy that can group DNA sequences into meaningful tokens.

Genomic sequences pose unique challenges for tokenization that differ from natural language, primarily due to the absence of a natural "word" unit. Prior works have largely adopted one of three approaches: single nucleotides (Dalla-Torre et al., 2025; Schiff et al., 2024), fixed-size k-mers (Poli et al., 2023; Ji et al., 2021), or Byte-Pair Encoding (BPE) (Zhou et al., 2024). The single nucleotide approach, while simple, results in excessively long sequences that make it computationally expensive and difficult to model long-range interactions (Dalla-Torre et al., 2025).

To circumvent this length issue, fixed-size k-mers and BPE have been explored, but these methods are inherently fixed and struggle to adapt to the biological context of DNA. K-mer tokenization is highly sensitive to small shifts, where a single insertion, deletion, or mutation can completely alter the tokenized output, even if the biological function remains unchanged (Dalla-Torre et al., 2025). Next, frequency-driven schemes like BPE fail to capture the functional importance of substrings, since the most frequent substrings are typically simple non-functional repetitive elements.

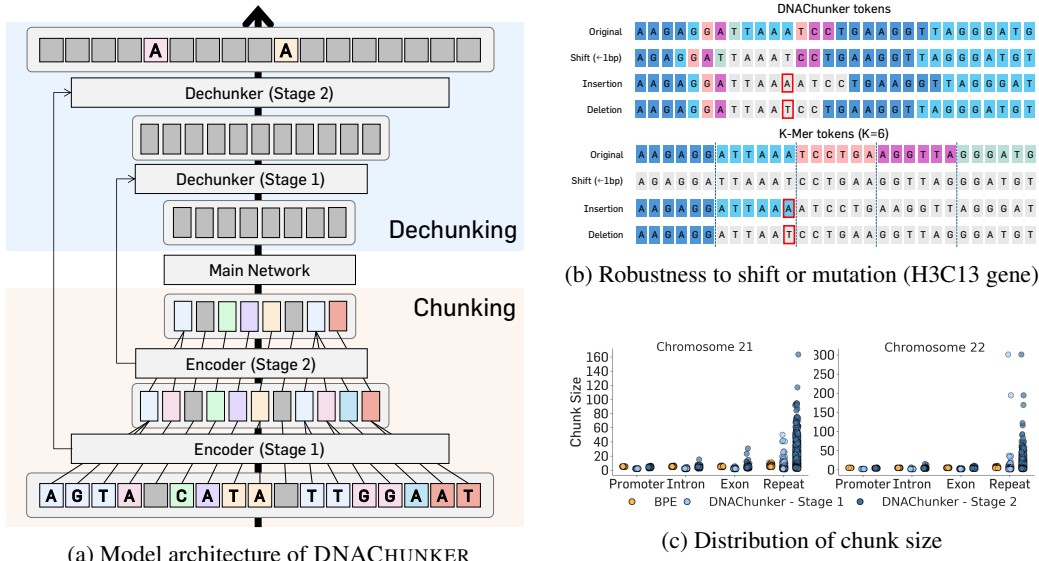

(a) Model architecture of DNACHUNKER

(b) Robustness to shift or mutation (H3C13 gene)

(c) Distribution of chunk size

Figure 1: **Architecture, tokenizer robustness, and distribution of chunk size.** (a) The architecture of DNACHUNKER. (b) Our tokenizer is robust against nucleotide-wise shifts or mutations, where we color the tokens to indicate that they are preserved despite the mutations. (c) Our DNACHUNKER dynamically represents functional elements (promoter, intron, exon) with high-resolution using smaller chunks, while compressing the non-functional repetitive elements with larger chunks.

To this end, we propose DNACHUNKER, a bidirectional masked DNA language model designed to overcome the limitations of fixed tokenization (Figure 1a). Our model leverages a learnable, dynamic tokenization mechanism proposed by Hwang et al. (2025) to group nucleotides into variable-length, biologically meaningful chunks directly from genomic data. We adopt a two-stage hierarchical architecture that processes raw input sequences with a lightweight bi-directional Caduceus (Schiff et al., 2024) layer and groups tokens based upon cosine similarity to form chunks. The representation is enhanced with an expressive main network, then subsequently upsampled back to the original base-pair resolution with a cross-attention based dechunking layer. Finally, a bi-directional Caduceus layer decodes the base pair-level representations to predict the masked nucleotide.

Importantly, in contrast to K-mer tokenization, our tokenizer is robust against nucleotide shifts, insertion, or mutation since the encoder is trained to chunk the raw sequence based on the context (Figure 1b). Furthermore, our dynamic chunking layers are trained to adaptively allocate chunk sizes to represent functional elements in high resolution, while compressing the non-functional repetitive elements with larger chunks (Figure 1c).

We validate DNACHUNKER by pre-training on the human reference genome (HG38) and fine-tuning on downstream tasks, namely Nucleotide Transformer (Dalla-Torre et al., 2025) and Genomic benchmarks (Grešová et al., 2023). DNACHUNKER achieves performance comparable to the state-of-the-art Generator (Wu et al., 2025) with 1.2B parameters, despite using significantly smaller 156M parameters. We also demonstrate that our dynamic tokenization considers biological context; We observe that our dynamic chunking procedure preserves crucial details through smaller, higher-resolution chunks for functional elements (regions with high phyloP scores, gene bodies, promoters, introns, exons), while assigning large chunks to non-functional elements (repetitive elements).

Overall, our contributions can be summarized as follows:

1. **Bidirectional masked DNA language model.** We adapt the dynamic chunking mechanism for the masked language model with bi-directional Caduceus and cross-attention layers (Section 3.1).
2. **State-of-the-art performance on various DNA benchmarks.** Our proposed DNACHUNKER outperforms the baselines in the Nucleotide Transformer and Genomic benchmarks (Section 4.1).
3. **Robustness and adaptivity of tokenization.** Our tokenization scheme is robust against mutations and adaptively allocates fine-grained representations for functional elements, while compressing non-functional elements with coarse-grained representations (Section 4.2).

## 2 RELATED WORKS

### 2.1 DNA LANGUAGE MODELS

**Autoregressive generation models.** While masked DNA LLMs excel at understanding and predicting DNA sequences, generative capabilities in this domain are still in their early stages. An early preprint on DNAGPT (Yang et al., 2024) demonstrated the ability to learn mammalian genomic structures through next-token prediction and other pre-training tasks. Recent works like HyenaDNA (Nguyen et al., 2023) and megaDNA (Shao & Yan, 2024) have achieved longer context lengths by employing the Hyena (Poli et al., 2023) and multiscale transformer architectures, respectively, though they are limited in their data and model scale. Next, Evo (Nguyen et al., 2024) was trained on an extensive dataset of prokaryotic and viral genomes. Evo2 (Brixi et al., 2025) extends this idea with 7B and 40B parameter models trained on 9.3 trillion DNA base pairs, achieving an unprecedented 1 million token context window with single-nucleotide resolution. Notably, Evo and Evo2 demonstrated practical utility by designing CRISPR-Cas molecular complexes (Nguyen et al., 2024) and bacteriophages (King et al., 2025) in the real world.

**Non-autoregressive generation models.** In addition, masked language models (MLMs) have been investigated for the representation learning of DNA sequences. MLMs are attractive since they allow reflection of the bidirectional nature of DNA sequences, e.g., regulatory motifs can act in both directions and functional prediction requires context from both upstream and downstream regions. The Nucleotide Transformer (NT; Dalla-Torre et al., 2025) scaled model parameters from 100 million to 2.5 billion and was trained on a diverse set of multispecies genomes. Subsequent studies, such as DNABERT-2 (Zhou et al., 2024) and GROVER (Sanabria et al., 2024) proposed to use Byte Pair Encoding (BPE) over k-mer tokenizers for masked DNA LLMs. A primary limitation of these models has been their insufficient context length, a consequence of the high computational cost of extending context in the standard transformer architecture. To address this, GENA-LM (Fishman et al., 2025) employs sparse attention, while Caduceus (Schiff et al., 2024) utilizes the lightweight BiMamba architecture (Tang et al., 2024).

### 2.2 LEARNABLE TOKENIZERS

**Autoregressive generation models.** Tokenization methods have primarily been developed in the context of autoregressive generation models. Existing models often rely on an outside function or module to identify boundaries. This includes delimiter-based methods like SpaceByte (Slagle, 2024), which works well for languages with clear separators, and entropy-based methods like the Byte Latent Transformer (Pagnoni et al., 2024), which identify boundaries based on conditional entropy. Recently, Hwang et al. (2025) proposed H-Net as a module to learn dynamic chunking, which learns optimal segmentation strategies directly from data through training and matches the performance of models based on fixed tokenizers for natural language and DNA tasks.

**Non-autoregressive generation models.** For non-autoregressive models, similar principles are applied with different design considerations. Charformer (Tay et al., 2022) introduced a gradient-based method for pooling sequences at multiple resolutions. Other approaches, such as eByte (Thawani et al., 2023) and Word-based self-attention fusion (Sreedhar et al., 2023), perform external chunking based on words. Our work, DNACHUNKER, fills a critical gap by being the first model to apply a learnable, dynamic chunking mechanism to a non-autoregressive, masked DNA language model. By adapting the core ideas of Hwang et al. (2025) to our architecture, we are able to eliminate the limitations of fixed tokenization and inefficient architectures.

## 3 METHODOLOGY

### 3.1 ARCHITECTURE DETAILS OF DNACHUNKER

DNACHUNKER is a masked language model (MLM) for genomic sequences designed around three modules: an encoder, a main network, and a decoder. The encoder compresses raw DNA sequences by grouping consecutive base pairs into coarse-grained chunks, enabling efficient downstream modeling. The main network then processes these chunked embeddings to capture long-range dependencies and contextual information across the genome. The decoder then restores base pair-level

resolution by upsampling the compressed chunks, allowing the model to predict masked nucleotides with high accuracy.

A key innovation of DNACHUNKER lies in its adaptation of the *dynamic chunking* algorithm, originally proposed in H-Net (Hwang et al., 2025) for autoregressive models, to the bidirectional framework of masked language modeling. This adaptation is supported by bidirectional encoders and decoders, which enable contextual information to flow in both forward and reverse directions along the sequence. Additionally, we employ a cross-attention mechanism between the encoder's pre-chunked embeddings and the decoder's outputs to further optimize the integration of multi-resolution information. This allows the model to leverage fine-grained uncompressed details for precise recovery of masked tokens informed by both upstream and downstream contexts. These components together improve the robustness and applicability of DNACHUNKER for a wide range of genomic downstream tasks. We provide an illustration of the architecture in Figure 1a and detailed hyperparameters in Section B in Appendix. In the following subsections, we provide details of the chunking and dechunking processes.

**Hierarchical chunking with dynamic boundaries.** The chunking process is designed to efficiently compress low-information regions in the DNA sequence while preserving high-information content at appropriate granularity. To achieve this, our encoder network employs a *two-stage* hierarchical chunking process that progressively transforms base pair–level signals into coarser, semantically meaningful representations. Each stage consists of three steps: (1) encoding the raw DNA sequence in the base pair-level embedding, (2) identifying decision boundaries between adjacent chunks, and (3) downsampling the embeddings according to these boundaries to produce the stage output. This structured process ensures that the model captures essential genomic patterns while reducing computational complexity. Both stages are implemented using the Caduceus architecture (Schiff et al., 2024), which efficiently models bidirectional dependencies. This architecture design is different from the original H-Net (Hwang et al., 2025), which employed a unidirectional encoder due to the autoregressive nature of its target task.

Formally, given an input sequence of length $T$, let $x^{(0)} = (x_1^{(0)}, \ldots, x_T^{(0)})$ denote the base pair-level embeddings. These embeddings are processed by the first stage encoder with Caduceus architecture, producing intermediate representations $\widehat{x}^{(0)}$. To adaptively segment the sequence into chunks, a separate routing network computes boundary probabilities $p_t^{(0)}$ for each position $t \in [1, T]$ using the cosine similarity between projected query and key vectors:

$$p_t^{(0)} = \frac{1}{2}\left(1 - \frac{(q_t^{(0)})^\top k_{t-1}^{(0)}}{\|q_t^{(0)}\| \cdot \|k_{t-1}^{(0)}\|}\right), \quad q_t^{(0)} = W_{\text{enc},q}^{(1)} \widehat{x}_t^{(0)}, \quad k_t^{(0)} = W_{\text{enc},k}^{(1)} \widehat{x}_t^{(0)}, \quad (1)$$

where $W_{\text{enc},q}^{(s)}$, $W_{\text{enc},k}^{(s)}$ are learnable parameters of the routing network of the encoder at stage $s \in \{1, 2\}$. The boundary indicator $b_t^{(s)}$ is obtained by thresholding the probability, i.e., $b_t^{(s)} = \mathbf{1}(p_t^{(s)} \geq 0.5)$. These indicators define chunk boundaries, allowing us to collect $T'$ adaptive chunks from $\widehat{x}^{(0)}$ where $T' = \sum_{t=1}^{T} b_t^{(0)}$. We denote the resulting chunked embeddings as $x^{(1)}$, and it is passed to the second encoder, which applies the same adaptive chunking process to create more coarse-grained representation $x^{(2)} = (x_1^{(2)}, ..., x_{T''}^{(2)})$ with $T'' < T'$. These embeddings $x^{(2)}$ then serve as input to the main network, which processes the compressed inputs to capture high-order dependencies.

Crucially, we implement a *masking protection mechanism* that enforces chunk boundaries before and after each masked base pair in the input. This ensures that masked tokens are protected and never merged into larger chunks, preventing the model from learning mask-specific tokenization patterns that would not generalize to downstream tasks without masked tokens.

**Main network.** The main network is composed of 8 Transformer blocks, where each one follows the standard Transformer architecture with layer normalization, multi-headed attention, and a feedforward network utilizing GELU activation. The attention mechanism incorporates Rotary Position Embeddings (RoPE; Su et al., 2024) to effectively encode positional information. The network accounts for the majority of parameters in DNACHUNKER and memory usage during inference, in contrast to the lightweight encoder and decoder components. However, by operating on compressed, chunked embeddings rather than raw DNA bases, the effective sequence length is substantially re-

duced. This design allows the Transformer to model long-range dependencies with significantly lower computational cost, while still retaining access to higher-order structural information.

**Hierarchical dechunking with cross-attention.**    Similarly to the chunking process, the decoder employs a two-stage hierarchical dechunking process to progressively expand compressed representations back to the full base-pair resolution. In contrast to chunking, dechunking relies on *cross-attention mechanism* with the intermediate chunked embeddings from the encoder. At each stage, the upsampling module uses encoder outputs as a query to guide the reconstruction of finer-grained representations. This design is inspired by U-Net architectures, where encoder features at multiple scales are reused to refine the decoder pathway. After the two dechunking stages, the reconstructed embeddings are passed through a single bidirectional Caduceus network, which differs from the encoder's two Caduceus models.

Specifically, let $z^{(0)} \in \mathbb{R}^{T'' \times d}$ be the output of the main network. The first dechunking stage produces the output $z^{(1)} \in \mathbb{R}^{T' \times d}$ using cross-attention between $z^{(0)}$ and $x^{(1)} \in \mathbb{R}^{T' \times d}$ (i.e., the first stage encoder's chunking embeddings) as

$$z^{(1)} = \text{softmax}\left( \frac{Q^{(1)} K^{(1)\top}}{\sqrt{d}} \right) V^{(1)}, \tag{2}$$

where $Q^{(1)} = x^{(1)} W^{(1)}_{\text{dec},q}$, $K^{(1)} = z^{(0)} W^{(1)}_{\text{dec},k}$, $V^{(1)} = z^{(0)} W^{(1)}_{\text{dec},v}$, and $d$ is the embedding dimension. Note that $W^{(s)}_{\text{dec},q}, W^{(1)}_{\text{dec},k}, W^{(s)}_{\text{dec},v}$ are learnable parameters in the routing network of decoder at stage $s \in \{1, 2\}$. The same process is applied to create upsampled embeddings $z^{(2)} \in \mathbb{R}^{T \times d}$ at the second stage dechunking.

Following the two-stage dechunking process, the resulting embeddings $z^{(2)}$ are combined with the first-stage encoder outputs $\widehat{x}^{(0)}$ via a residual connection, enhancing detail retention. The reconstructed representation then serves as an input to the final decoder network, which employs a bidirectional Caduceus model (Schiff et al., 2024), producing logits for accurate masked nucleotide prediction. This final processing ensures that the model's predictions maintain biological coherence while benefiting from the multi-scale contextual information captured throughout the hierarchical encoding-processing-decoding pipeline. More details on architecture are provided in Section B.

### 3.2 Model Pretraining

**Loss function.**    DNACHUNKER is pretrained with masked language modeling, with down-weighting of repetitive regions of DNA by 0.1, in-line with prior works (Brixi et al., 2025). The loss is formulated as follows:

$$\mathcal{L}_{\text{MLM}} = \sum_{t \in M} w_t \mathcal{L}_{\text{CE}}(t) \quad w_t = \begin{cases} 0.1 & \text{if position } t \text{ is in a repetitive region} \\ 1.0 & \text{otherwise} \end{cases}, \tag{3}$$

where $\mathcal{L}_{\text{CE}}(t)$ denotes the cross entropy loss for predicting the masked nucleotide at position $t$. Additionally, to control the degree of compression from the chunking layers, we use the ratio loss proposed by Hwang et al. (2025) at stage $s \in \{0, 1\}$:

$$\mathcal{L}^{(s)}_{\text{ratio}} = \frac{\overline{b}^{(s)} \overline{p}^{(s)}}{\alpha^{(s)}} + \frac{(1 - \overline{b}^{(s)})(1 - \overline{p}^{(s)})}{1 - \alpha^{(s)}}, \quad \overline{b}^{(s)} = \frac{1}{T} \sum_{t=1}^{T} b_t^{(s)}, \quad \overline{p}^{(s)} = \frac{1}{T} \sum_{t=1}^{T} p_t^{(s)}, \tag{4}$$

where $\overline{b}^{(s)}$ and $\overline{p}^{(s)}$ are the fraction of selected tokens and the average boundary probability, respectively, and $\alpha^{(s)} \in (0, 1)$ is the target compression ratio of the encoder, which is a controllable parameter. Note that $\overline{b}^{(s)}$ is non-differentiable, but the network can be trained towards the target compression ratio through tuning $\overline{p}^{(s)}$. Together, we train the model to minimize the loss $\mathcal{L} = \mathcal{L}_{\text{MLM}} + \lambda \mathcal{L}^{(0)}_{\text{ratio}} + \lambda \mathcal{L}^{(1)}_{\text{ratio}}$. More details about pretraining can be found in Section A.

**Dataset.**    We pretrain our model on the Human Reference Genome, adopting the data partitioning strategy from Enformer (Avsec et al., 2021). The genome is first divided into non-overlapping regions of $2^{20}$ (1,048,576) base pairs (bp), which will be allocated to the training, validation, and

test sets. These regions are subsequently segmented into input sequences with a maximum length of 2048 bp. During the preprocessing, ambiguous nucleotides ('N') are mapped to a padding token and are excluded from the loss computation. Following the methodology of BERT (Devlin et al., 2019), for each input sequence, 15% of all nucleotides are randomly selected for prediction. Of this selection, 80% are replaced with a [MASK] token, 10% are substituted with a random nucleotide, and the remaining 10% are left unchanged.

**Fine-tuning on downstream tasks.** For fine-tuning on the downstream tasks, we remove the language model head and perform average pooling over the valid tokens, *i.e.* excluding [PAD] tokens. The pooled output is subsequently passed through a linear layer.

## 4 EXPERIMENTS

In what follows, we demonstrate the experimental results for evaluating DNACHUNKER upon two benchmark datasets: Nucleotide Transformer benchmark (Dalla-Torre et al., 2025) and Genomic benchmark (Grešová et al., 2023). We show that, despite the small number of parameters (156M), DNACHUNKER demonstrates state-of-the-art performance (Section 4.1). Next, we describe ablative experiments comparing vanilla H-Net with DNACHUNKER, and provide extensive analysis of the learned tokenizer, demonstrating its robustness and inherent biological understanding (Section 4.2).

### 4.1 DOWNSTREAM TASKS

**Nucleotide Transformer benchmark.** We evaluate our model on the Nucleotide Transformer benchmark (Dalla-Torre et al., 2025), which aggregates 18 datasets spanning three task families: (i) *histone mark prediction* from chromatin profiling assays, (ii) *regulatory annotation* such as promoter and enhancer classification, and (iii) *splice-site* annotation at donor/acceptor boundaries. Following the evaluation protocol of Wu et al. (2025), we perform 10-fold cross-validation and report the Matthews Correlation Coefficient (MCC) for each dataset and the average rank among 10 models. Specific finetuning details of DNACHUNKER are reported in C.1, while scores of previous baseline models are taken from Wu et al. (2025).

Results are summarized in Table 1. DNACHUNKER achieves state-of-the-art performance on 10 out of 18 datasets, bypassing the previous best model, Generator (Wu et al., 2025), by a large margin in both average MCC and average rank. Our gains are especially more pronounced upon the histone mark prediction tasks, showing an average gain of 14.2%. Note that DNACHUNKER is trained only on the human reference genome, using only 13% of the Generator's number of parameters.

**Genomic benchmark.** The Genomic benchmark suite (Grešová et al., 2023) consists of eight binary regulatory-element classification tasks over short to mid DNA windows (approximately 200-2000 bp), covering *enhancer* and *promoter* recognition, *coding vs. intergenic* discrimination, and a small *species* control (human vs. worm). We follow the evaluation protocol of Schiff et al. (2024) and report the top-1 accuracy averaged over 5-fold cross-validation. Since the train/test splits used in Schiff et al. (2024) differ from those in Wu et al. (2025), we reproduce the Generator results by fine-tuning the pretrained model on the splits selected by Schiff et al. (2024). Specific finetuning details are reported in C.2, while scores from baseline models are taken from Schiff et al. (2024).

Our DNACHUNKER achieves the best average rank and the second-highest average accuracy while using 7.69× fewer parameters than the best model Generator (Wu et al., 2025) with 1.2B parameters. Additionally, our model was trained solely upon the human reference genome, while Generator used various types of genomes beyond human. This reflects the modest performance among species-related tasks such as Mouse Enhancers. Interestingly, Generator exhibits significant fluctuation in performance, showing exceptionally lower ranks in tasks such as Human NonTATA Promoters and Human OCR Ensemble. In contrast, DNACHUNKER consistently exhibits good performance, evidenced by the best average rank.

### 4.2 ABLATIVE STUDIES

Table 1: **Nucleotide Transformer Benchmark.** The reported values represent the Matthews Correlation Coefficient (MCC; mean ± standard error) averaged over 10-fold cross-validation. Best results are **bold**; second best are underlined.

| | Enformer (252M) | DNABERT-2 (117M) | HyenaDNA (55M) | NT-multi (2.5B) | NT-v2 (500M) | Caduceus-Ph (8M) | Caduceus-PS (8M) | GROVER (87M) | Generator (1.2B) | DNACHUNKER (156M) |
|---|---|---|---|---|---|---|---|---|---|---|
| ***Histone Markers*** | | | | | | | | | | |
| H3 | 0.724 ± 0.018 | 0.785 ± 0.012 | 0.781 ± 0.015 | 0.793 ± 0.013 | 0.788 ± 0.010 | 0.794 ± 0.012 | 0.772 ± 0.022 | 0.768 ± 0.008 | 0.806 ± 0.005 | **0.827** ± 0.008 |
| H3K14ac | 0.284 ± 0.024 | 0.515 ± 0.009 | 0.608 ± 0.020 | 0.538 ± 0.009 | 0.538 ± 0.015 | 0.564 ± 0.033 | 0.596 ± 0.038 | 0.548 ± 0.020 | 0.605 ± 0.008 | **0.710** ± 0.022 |
| H3K36me3 | 0.345 ± 0.019 | 0.591 ± 0.005 | 0.614 ± 0.014 | 0.618 ± 0.011 | 0.618 ± 0.015 | 0.590 ± 0.018 | 0.611 ± 0.048 | 0.563 ± 0.017 | 0.657 ± 0.007 | **0.671** ± 0.003 |
| H3K4me1 | 0.291 ± 0.016 | 0.512 ± 0.008 | 0.512 ± 0.008 | 0.541 ± 0.005 | 0.544 ± 0.009 | 0.468 ± 0.015 | 0.487 ± 0.029 | 0.461 ± 0.018 | 0.553 ± 0.009 | **0.621** ± 0.010 |
| H3K4me2 | 0.207 ± 0.021 | 0.333 ± 0.013 | 0.455 ± 0.028 | 0.324 ± 0.014 | 0.302 ± 0.020 | 0.332 ± 0.034 | 0.431 ± 0.016 | 0.403 ± 0.042 | 0.424 ± 0.013 | **0.596** ± 0.024 |
| H3K4me3 | 0.156 ± 0.022 | 0.353 ± 0.021 | 0.550 ± 0.015 | 0.408 ± 0.011 | 0.437 ± 0.028 | 0.490 ± 0.042 | 0.528 ± 0.033 | 0.458 ± 0.022 | 0.512 ± 0.009 | **0.659** ± 0.047 |
| H3K79me3 | 0.498 ± 0.013 | 0.615 ± 0.010 | 0.669 ± 0.014 | 0.623 ± 0.014 | 0.621 ± 0.012 | 0.641 ± 0.028 | 0.682 ± 0.018 | 0.626 ± 0.026 | 0.670 ± 0.011 | **0.751** ± 0.022 |
| H3K9ac | 0.415 ± 0.020 | 0.545 ± 0.009 | 0.586 ± 0.021 | 0.547 ± 0.011 | 0.567 ± 0.020 | 0.575 ± 0.024 | 0.564 ± 0.018 | 0.581 ± 0.015 | 0.612 ± 0.006 | **0.678** ± 0.011 |
| H4 | 0.735 ± 0.023 | 0.797 ± 0.008 | 0.763 ± 0.012 | 0.808 ± 0.007 | 0.795 ± 0.008 | 0.788 ± 0.010 | 0.799 ± 0.010 | 0.769 ± 0.017 | **0.815** ± 0.008 | 0.812 ± 0.011 |
| H4ac | 0.275 ± 0.022 | 0.465 ± 0.013 | 0.564 ± 0.011 | 0.492 ± 0.014 | 0.502 ± 0.025 | 0.548 ± 0.027 | 0.585 ± 0.018 | 0.530 ± 0.017 | 0.592 ± 0.015 | **0.687** ± 0.027 |
| **Average MCC (↑)** | 0.393 | 0.551 | 0.610 | 0.569 | 0.571 | 0.579 | 0.606 | 0.571 | 0.625 | **0.701** |
| ***Regulatory Annotation*** | | | | | | | | | | |
| Enhancer | 0.454 ± 0.029 | 0.525 ± 0.026 | 0.520 ± 0.031 | 0.545 ± 0.028 | 0.561 ± 0.029 | 0.522 ± 0.024 | 0.511 ± 0.026 | 0.516 ± 0.018 | **0.580** ± 0.015 | 0.556 ± 0.021 |
| Enhancer Type | 0.312 ± 0.043 | 0.423 ± 0.018 | 0.403 ± 0.056 | 0.444 ± 0.022 | 0.444 ± 0.036 | 0.403 ± 0.028 | 0.410 ± 0.026 | 0.433 ± 0.029 | 0.477 ± 0.017 | **0.521** ± 0.022 |
| Promoter All | 0.910 ± 0.004 | 0.945 ± 0.003 | 0.919 ± 0.003 | 0.951 ± 0.004 | 0.952 ± 0.002 | 0.937 ± 0.002 | 0.941 ± 0.003 | 0.926 ± 0.004 | 0.962 ± 0.002 | **0.968** ± 0.011 |
| Promoter NonTATA | 0.910 ± 0.006 | 0.944 ± 0.003 | 0.919 ± 0.004 | 0.969 ± 0.004 | 0.952 ± 0.003 | 0.935 ± 0.007 | 0.940 ± 0.002 | 0.925 ± 0.004 | 0.962 ± 0.001 | **0.969** ± 0.011 |
| Promoter TATA | 0.920 ± 0.012 | 0.911 ± 0.011 | 0.881 ± 0.020 | 0.919 ± 0.008 | 0.933 ± 0.009 | 0.895 ± 0.010 | 0.903 ± 0.010 | 0.891 ± 0.009 | 0.948 ± 0.008 | **0.965** ± 0.005 |
| **Average MCC (↑)** | 0.701 | 0.750 | 0.728 | 0.766 | 0.768 | 0.738 | 0.741 | 0.738 | 0.786 | **0.796** |
| ***Splice Site Annotation*** | | | | | | | | | | |
| Splice Acceptor | 0.772 ± 0.007 | 0.909 ± 0.004 | 0.935 ± 0.005 | 0.973 ± 0.002 | 0.973 ± 0.004 | 0.918 ± 0.017 | 0.907 ± 0.015 | 0.912 ± 0.010 | **0.981** ± 0.002 | 0.968 ± 0.011 |
| Splice Site All | 0.831 ± 0.012 | 0.950 ± 0.003 | 0.917 ± 0.006 | 0.974 ± 0.004 | 0.975 ± 0.002 | 0.935 ± 0.011 | 0.953 ± 0.005 | 0.919 ± 0.005 | **0.978** ± 0.001 | 0.968 ± 0.030 |
| Splice Donor | 0.813 ± 0.015 | 0.927 ± 0.003 | 0.894 ± 0.013 | 0.974 ± 0.004 | 0.977 ± 0.007 | 0.912 ± 0.009 | 0.930 ± 0.010 | 0.888 ± 0.012 | **0.978** ± 0.002 | 0.960 ± 0.011 |
| **Average MCC (↑)** | 0.805 | 0.929 | 0.915 | 0.974 | 0.975 | 0.922 | 0.930 | 0.906 | **0.979** | 0.965 |
| **Total Average MCC (↑)** | 0.547 | 0.669 | 0.694 | 0.690 | 0.693 | 0.680 | 0.697 | 0.673 | 0.728 | **0.772** |
| **Total Average Rank (↓)** | 9.67 | 6.72 | 6.00 | 4.83 | 4.56 | 6.33 | 5.61 | 7.22 | 2.06 | **1.67** |

Table 2: **Genomic Benchmarks.** The reported values represent accuracy (mean ± standard error) averaged over 5-fold cross-validation. Best results are **bold**; second best are underlined.

| | CNN (264k) | HyenaDNA (436k) | Mamba (468k) | Caduceus-Ph (470k) | Caduceus-PS (470k) | Generator (1.2B) | DNACHUNKER (156M) |
|---|---|---|---|---|---|---|---|
| Mouse Enhancers | 0.715 ± 0.087 | 0.780 ± 0.025 | 0.743 ± 0.054 | 0.754 ± 0.074 | 0.793 ± 0.058 | **0.853** ± 0.018 | 0.833 ± 0.016 |
| Coding vs. Intergenomic | 0.892 ± 0.008 | 0.904 ± 0.005 | 0.904 ± 0.004 | 0.915 ± 0.003 | 0.910 ± 0.003 | **0.960** ± 0.001 | 0.926 ± 0.002 |
| Human vs. Worm | 0.942 ± 0.002 | 0.964 ± 0.002 | 0.967 ± 0.002 | 0.973 ± 0.001 | 0.968 ± 0.002 | **0.979** ± 0.001 | 0.969 ± 0.001 |
| Human Enhancers Cohn | 0.702 ± 0.021 | 0.729 ± 0.014 | 0.732 ± 0.029 | **0.747** ± 0.004 | 0.745 ± 0.007 | 0.759 ± 0.002 | 0.744 ± 0.005 |
| Human Enhancer Ensembl | 0.744 ± 0.122 | 0.849 ± 0.006 | 0.862 ± 0.008 | 0.893 ± 0.008 | 0.900 ± 0.006 | 0.877 ± 0.007 | **0.906** ± 0.002 |
| Human Regulatory | 0.872 ± 0.005 | 0.869 ± 0.012 | 0.814 ± 0.211 | 0.872 ± 0.011 | 0.873 ± 0.007 | **0.930** ± 0.000 | 0.880 ± 0.011 |
| Human OCR Ensembl | 0.698 ± 0.013 | 0.783 ± 0.007 | 0.815 ± 0.002 | **0.828** ± 0.006 | 0.818 ± 0.006 | 0.816 ± 0.004 | 0.818 ± 0.004 |
| Human NonTATA Promoters | 0.861 ± 0.009 | 0.944 ± 0.002 | 0.933 ± 0.007 | 0.946 ± 0.007 | 0.945 ± 0.010 | 0.925 ± 0.007 | **0.957** ± 0.09 |
| **Average Accuracy (↑)** | 0.803 | 0.853 | 0.846 | 0.866 | 0.869 | **0.887** | 0.879 |
| **Average Rank (↓)** | 6.75 | 5.44 | 5.31 | 2.75 | 3.06 | 2.50 | **2.19** |

**Architectural contributions.** In Figure 2, we compare the results of DNACHUNKER against H-Net architecture (Hwang et al., 2025) to validate the contribution of the architecture proposed in our work. To this end, we compare with H-Net trained using the same masked language model scheme on the same human reference genome. Note that our model incorporates two key architectural improvements over the vanilla H-Net: (1) the pass-through of special tokens to the main model and (2) a cross-attention-based dechunking scheme that replaces the original smoothing module.

As shown in Figure 2, DNACHUNKER exhibits superior performance across all tasks, with particularly notable gains on the splice site annotation tasks. This outcome underscores the critical role of the cross-attention dechunking scheme, which effectively models bidirectionality, a capability the vanilla H-Net lacks, due to its inherent design of the smoothing module. These results collectively

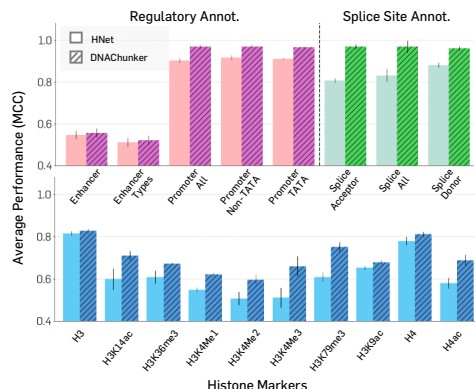

Figure 2: **Comparison with H-Net upon NT Benchmark.** DNACHUNKER compared with H-Net trained with masked language modeling. Results are averaged over 10-fold cross validations with error bars.

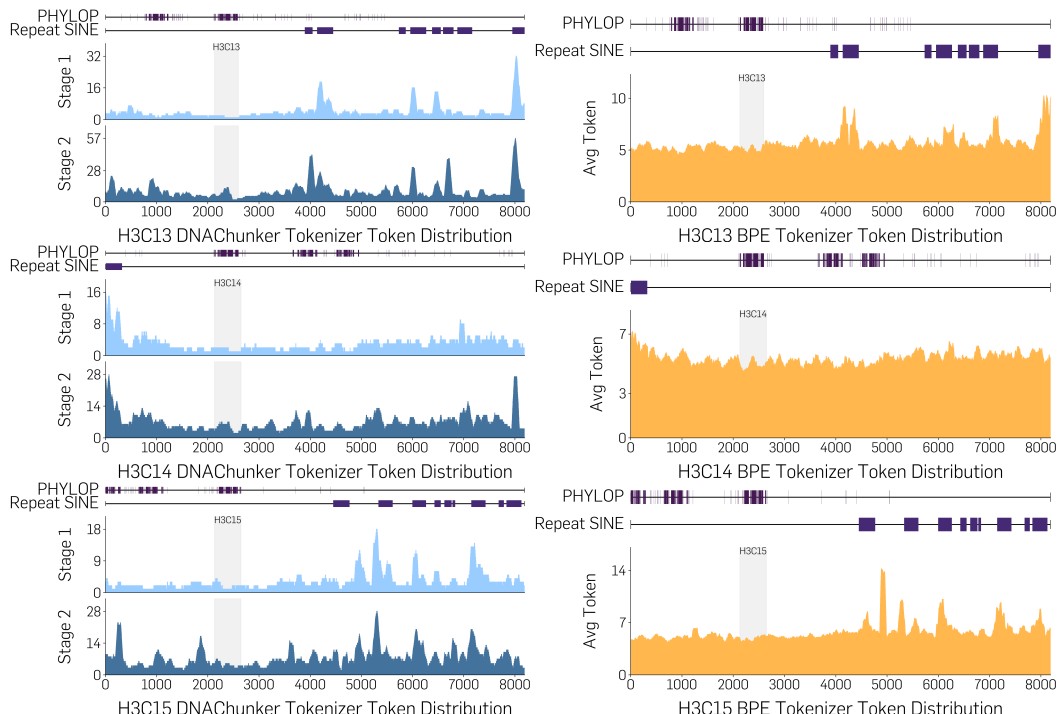

Figure 3: **Token size distributions of BPE and DNACHUNKER.** The BPE tokenizer (right) is compared against the two-stage DNACHUNKER tokenizer (left) on the H3C13, H3C14, and H3C15 genes. The plots visualize the average token size of BPE and the Stage 1 and Stage 2 token sizes of DNACHUNKER. Key genomic features are included as a reference, like gene bodies (shaded regions), conserved elements (PHYLOP), and SINE repeats.

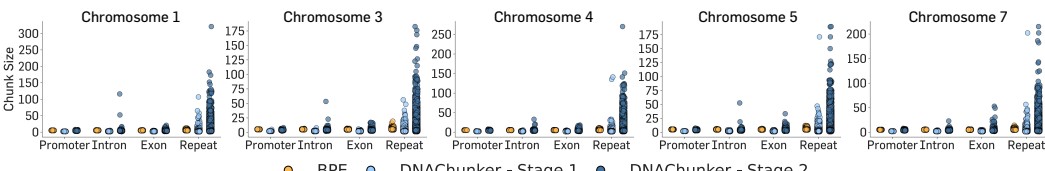

Figure 4: **Token size distributions of BPE and DNACHUNKER.** A comparison of the BPE tokenizer against our two-stage DNACHUNKER tokenizer on human chromosomes 1, 3, 4, 5, and 7. The plots visualize the distribution of chunk sizes for BPE and both the stage 1 and stage 2 outputs of DNACHUNKER. The distributions are categorized by key genomic features including Promoter, Intron, Exon, and Repeat.

demonstrate the necessity of our architectural modifications for applying H-Net to genomic sequences and validate the efficacy of our proposed approach.

**Token size distribution.** To qualitatively assess the tokenization strategy of DNACHUNKER, we investigate its token size distribution against a fixed BPE tokenizer from DNABERT-2 (Zhou et al., 2024) across specific genomic loci. In particular, we select three genes from the histone cluster 1 H3 family, i.e., H3C13, H3C14, and H3C15, that feature both highly-conserved coding sequences with high phyloP scores (PHYLOP) and repetitive Short Interspersed Nuclear Elements (Repeat SINE). Ideally, the tokenizer should represent the gene bodies and the PHYLOP regions with high resolution using small chunks, while compressing the redundant Repeat SINE region with large chunks.

Figure 3 illustrates the results, highlighting a stark contrast between the two methods. The right column (orange color) shows that the BPE tokenizer's distribution is largely uniform, applying a consistent token granularity irrespective of the underlying biological annotations. It fails to differentiate between the highly conserved regions, indicated by high PhyloP scores, and less informative

Table 3: **Robustness of tokenizers against mutations.** Similarity scores (mean $\pm$ standard error) calculated between the tokenized results of reference sequence and mutated sequence, from sampled ClinVar dataset. High scores indicate the tokenizers' robustness to mutations.

| | SNV | | InDel | |
| --- | --- | --- | --- | --- |
| | Benign | Pathogenic | Benign | Pathogenic |
| BPE Tokenizer | $0.9993 \pm 0.0000$ | $0.9994 \pm 0.0006$ | $0.7506 \pm 0.0462$ | $0.7434 \pm 0.0325$ |
| DNACHUNKER (Stage 1) | $0.9987 \pm 0.0005$ | $0.9985 \pm 0.0005$ | $0.8512 \pm 0.0223$ | $0.8492 \pm 0.16$ |
| DNACHUNKER (Stage 2) | $0.9940 \pm 0.0020$ | $0.9934 \pm 0.0021$ | $0.7932 \pm 0.0296$ | $0.7900 \pm 0.0236$ |

repetitive SINE elements. On the other hand, the left column (blue color) reveals that DNACHUNKER adapts its dynamic chunking strategy to the biological context. In regions annotated as SINEs, characterized by low sequence complexity and less functional significance, DNACHUNKER allocates larger chunks, effectively compressing this redundant information. Conversely, for regions of high evolutionary conservation and within core gene bodies, the tokenizer employs finer chunks.

In Figures 1c and 4, we additionally analyze the chunk size distribution across annotated genomic regions (Promoters, Introns, Exons, and Repeats) on a diverse set of chromosomes: 1, 3, 4, 5, 7, 21, and 22. Our method produced highly variable chunk sizes sensitive to the local genomic context, with some chunks in repeat regions reaching up to 320 base pairs. In contrast, standard BPE tokenization generates chunks of mostly uniform length, regardless of the underlying biological information. This adaptive behavior indicates that our model, despite being trained solely on a masked language modeling objective, learns to parse the genome in a manner that reflects its inherent biological architecture. This increased resolution in critical areas allows for more computational resources to be designated to regions dense with biological information.

**Robustness to mutations.** To quantitatively evaluate the stability of our learnable tokenization, we benchmarked its performance against standard BPE tokenizer in the genetic variants sampled from the ClinVar (Landrum et al., 2016) dataset. Specifically we take 1,000 samples for each type: Benign single nucleotide variants (SNVs) and InDels, along with Pathogenic SNVs and InDels. To quantify the robustness, we introduce a similarity metric to measure the similarity between two tokenized outputs: $S(x^{\text{ref}}, x^{\text{mut}}) = (1 - \gamma)S_{\text{boundary}} + \gamma S_{\text{content}}$ where $x^{\text{ref}}$ and $x^{\text{mut}}$ denote the tokenization of the reference and the mutated sequences, $S_{\text{boundary}}$ is the Jaccard Similarity of the boundaries formed by tokenization protocols, and $S_{\text{content}}$ denote the edit distance between two tokenizations. Ideally, $S_{\text{boundary}}$ captures the structural similarity of how the tokenizer divides the input sequence, whereas $S_{\text{content}}$ captures the content similarity between the two. For our experiments, we choose $\gamma = 0.5$. We present the results in Table 3. Both BPE and our tokenization demonstrates relatively high similarity in simple SNVs in both benign and pathogenic mutations. For insertions and deletions, we find DNACHUNKER to achieve higher robustness scores in both benign and pathogenic mutations.

## 5 CONCLUSION

In this work, we address the fundamental challenge of tokenization in genomic language models: the absence of natural semantic units analogous to words in human language. This complicates the development of biologically meaningful tokenization strategies, necessitating the need for a data-driven approach. To this end, we propose DNACHUNKER, leveraging a learnable, dynamic tokenization strategy designed for genomic language modeling. Our extensive experiments show that DNACHUNKER consistently outperforms prior baselines across benchmark datasets. Furthermore, our ablative studies reveal that the model's learned tokenization is *not* arbitrary but biologically informed; assigning smaller, higher-resolution tokens for functional elements while assigning larger chunks to redundant sections. Ultimately, these results underscore the effectiveness of employing a learnable tokenization strategy for more biologically aware genomic language models.

## ETHICS STATEMENT

The genomic data used for pre-training our model is the Human Reference Genome (HG38), which is a publicly available and fully anonymized resource widely used by the international scientific community. The use of this public reference genome ensures that our work does not involve private or identifiable genetic information, thereby posing no direct risk to individual privacy. While our research focuses on developing a foundational language model for understanding DNA sequences, we acknowledge that powerful genomic models could have future applications with broader ethical implications.

## REPRODUCIBILITY STATEMENT

To ensure the reproducibility of our results, we have provided a detailed account of our methodology and experimental setup. Our model was pre-trained on the Human Reference Genome (HG38), using the public data partitioning strategy from the Enformer study. For downstream evaluation, we used two publicly available collections: the Nucleotide Transformer benchmark and the Genomic benchmark. Comprehensive details regarding the model architecture, pre-training configuration, and fine-tuning hyperparameters for every task are documented in the Appendices. We provide our source code at https://anonymous.4open.science/r/DNAChunker_final-7FD6/ for reproducibility with an appropriate open-source license.

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

# A  PRE-TRAINING DETAILS

Table 4 summarizes the pretraining setup of DNACHUNKER, including dataset specifications, optimization strategy, and masking details. The model is trained on the Enformer study splits, covering 34,021 genomic segments with a maximum sequence length of $2^{13}$ (8,192 bp), amounting to approximately 35 billion base pairs in total. To ensure scalability across different context lengths, we adopt a constant token budget of $2^{20}$ tokens per batch. This results in dynamically adjusted batch sizes depending on the sequence length: for instance, sequences of length 1,024 are processed in batches of 1,024, whereas long sequences of length $2^{17}$ (131k) are trained with a reduced batch size of 8.

Optimization is performed with the Adam optimizer (Kingma, 2015), using $\beta_1 = 0.95$ and $\beta_2 = 0.9$, and a learning rate of $5 \times 10^{-4}$ following a cosine decay schedule. Pretraining follows a masked language modeling objective with 15% of input tokens selected for corruption: 80% of these are replaced with a [MASK] token, 10% with a random base, and 10% left unchanged. This bi-directional masking scheme encourages the model to leverage both local and global dependencies within DNA sequences while learning robust, context-aware representations.

Table 4: Pre-training Hyperparameters and Dataset Details

| Component | Hyperparameter | Value |
|---|---|---|
| Dataset | Source | Enformer study splits |
| | Training Segments | 34,021 |
| | Max Sequence Length | 8192 ($2^{13}$) |
| | Total Tokens | $\approx$ 35 billion base pairs |
| Training Configuration | Tokens per Batch | $2^{20}$ (constant) |
| | Example Batch Sizes | Seq length 1,024 $\rightarrow$ Batch size 1,024; Seq length 131k ($2^{17}$) $\rightarrow$ Batch size 8 |
| Optimizer & Learning Rate | Optimizer | ADAM (Kingma, 2015) |
| | ADAM $\beta_1$ | 0.95 |
| | ADAM $\beta_2$ | 0.9 |
| | LR | $5 \times 10^{-4}$ |
| | LR Schedule | Cosine Decay |
| Masking (bi-directional) | Masking Percentage | 15% of input tokens |
| | Masking Strategy | 80% replaced with [MASK] token, 10% replaced with a random token, and 10% unchanged |

# B  ARCHITECTURE DETAILS

Table 5 summarizes the architectural configuration of DNACHUNKER, which comprises 156M parameters in total. The model follows a hierarchical encoder–decoder design with routing modules and cross-attention upsamplers. The encoder is structured in two stages, each consisting of a lightweight 2-layer BiMamba (Caduceus) backbone paired with a learnable routing module that projects query and key representations in a 1024-dimensional space. These hierarchical encoders progressively compress the input representation before passing it to the main network.

The main processing block of DNACHUNKER is an 8-layer Transformer stack (100M parameters), employing rotary position embeddings (RoPE) for attention, 8 heads with 128 dimensions each, RMSNorm applied to query/key projections, and Pre-LayerNorm across both attention and MLP layers. The MLPs expand the embedding dimension from 1024 to 4096 using GELU activations. On the decoder side, a single 2-layer BiMamba is coupled with two cross-attention upsamplers, which reintroduce fine-grained information from the encoder through learned projection matrices. A residual projection layer and a final RMSNorm complete the architecture. Together, these carefully balanced components enable DNACHUNKER to achieve high representational capacity while maintaining efficiency.

Table 5: Hyperparameters of DNACHUNKER architecture (156M parameters in total).

| Component | Architecture / Details | #Params |
|---|---|---|
| Token embedding | 16 vocab size, 1024 dim | – |
| Encoder (Stage 1) | 2-layer BiMamba (Caduceus) | 14M |
| Router (Stage 1) | Routing module ($W_{\text{enc},q}^{(1)}, W_{\text{enc},k}^{(1)} \in \mathbb{R}^{1024 \times 1024}$) | 2M |
| Encoder (Stage 2) | 2-layer BiMamba (Caduceus) | 14M |
| Router (Stage 2) | Routing module ($W_{\text{enc},q}^{(2)}, W_{\text{enc},k}^{(2)} \in \mathbb{R}^{1024 \times 1024}$) | 2M |
| Main network | 8-layer Transformer blocks
• Attention: RoPE, 8 heads, 128 dim per head, RMSNorm for query/key
• MLP: 1024 input dim, 4096 hidden dim, GELU
• Pre-LayerNorm for Attention and MLP | 100M |
| Decoder | 2-layer BiMamba (Caduceus) | 14M |
| Upsampler 1 | Cross-attention upsampler
• $W_{\text{dec},q}^{(1)}, W_{\text{dec},k}^{(1)}, W_{\text{dec},v}^{(1)}, W_{\text{dec},o}^{(1)} \in \mathbb{R}^{1024 \times 1024}$ | 4M |
| Upsampler 2 | Cross-attention upsampler
• $W_{\text{dec},q}^{(2)}, W_{\text{dec},k}^{(2)}, W_{\text{dec},v}^{(2)}, W_{\text{dec},o}^{(2)} \in \mathbb{R}^{1024 \times 1024}$ | 4M |
| Residual Projection | Linear(1024 → 1024) | 1M |
| Final Normalization | RMSNorm | – |

## C  FINETUNING DETAILS ON DOWNSTREAM TASKS

### C.1  NUCLEOTIDE TRANSFORMER BENCHMARK

We fine-tune DNACHUNKER with a search space over learning rates $\{1 \times 10^{-5}, 5 \times 10^{-5}, 1 \times 10^{-4}\}$ and effective batch sizes $\{32, 64\}$. We use attention pooling over token embeddings and *do not* apply RC augmentation or conjoining. Training runs for up to 15 epochs with shuffling each epoch; the best-validation checkpoint is used for scoring. Hyperparameters for the DNACHUNKER reported in Table 1 can be found in Table 6. All experiments use a single NVIDIA A100 GPU with 40GB VRAM.

Table 6: Hyperparameter settings for DNACHUNKER on Nucleotide Transformer benchmark.

| Task | LR | BS |
|------|----|----|
| H3 | $5 \times 10^{-5}$ | 32 |
| H3K14ac | $5 \times 10^{-5}$ | 32 |
| H3K36me3 | $5 \times 10^{-5}$ | 32 |
| H3K4me1 | $5 \times 10^{-5}$ | 32 |
| H3K4me2 | $5 \times 10^{-5}$ | 32 |
| H3K4me3 | $5 \times 10^{-5}$ | 32 |
| H3K79me3 | $5 \times 10^{-5}$ | 32 |
| H3K9ac | $5 \times 10^{-5}$ | 32 |
| H4 | $1 \times 10^{-4}$ | 32 |
| H4ac | $5 \times 10^{-5}$ | 32 |
| Enhancers | $5 \times 10^{-5}$ | 32 |
| Enhancers types | $5 \times 10^{-5}$ | 32 |
| Promoter all | $5 \times 10^{-5}$ | 32 |
| Promoter non-TATA | $1 \times 10^{-4}$ | 32 |
| Promoter TATA | $5 \times 10^{-5}$ | 64 |
| Splice sites acceptors | $1 \times 10^{-4}$ | 32 |
| Splice sites all | $1 \times 10^{-4}$ | 32 |
| Splice sites donors | $1 \times 10^{-4}$ | 32 |

## C.2 GENOMICS BENCHMARK

We fine-tune DNACHUNKER and Generator with a search space over learning rates between $(5 \times 10^{-6}, 1 \times 10^{-4})$ and effective batch sizes between $\{16, 128\}$. We use average pooling over token embeddings and *do not* apply RC augmentation or conjoining. Training runs for up to 20 epochs with shuffling each epoch; the best-validation checkpoint is used for scoring. Hyperparameters for the DNACHUNKER reported in Table 2 can be found in Table 7. All experiments use a single NVIDIA A100 GPU with 40GB VRAM.

Table 7: Hyperparameter settings for Generator and DNACHUNKER on Genomic benchmark.

|  | Generator | | DNACHUNKER | |
| --- | --- | --- | --- | --- |
|  | LR | BS | LR | BS |
| Mouse Enhancers | $5 \times 10^{-5}$ | 4 | $5 \times 10^{-6}$ | 16 |
| Coding vs. Intergenomic | $2 \times 10^{-5}$ | 8 | $5 \times 10^{-6}$ | 32 |
| Human vs. Worm | $2 \times 10^{-5}$ | 8 | $5 \times 10^{-6}$ | 32 |
| Human Enhancers Cohn | $1 \times 10^{-5}$ | 8 | $5 \times 10^{-6}$ | 32 |
| Human Enhancer Ensembl | $5 \times 10^{-5}$ | 32 | $1 \times 10^{-5}$ | 32 |
| Human Regulatory | $1 \times 10^{-5}$ | 8 | $5 \times 10^{-4}$ | 64 |
| Human OCR Ensembl | $1 \times 10^{-5}$ | 8 | $5 \times 10^{-4}$ | 64 |
| Human NonTATA Promoters | $5 \times 10^{-5}$ | 8 | $1 \times 10^{-4}$ | 128 |

