# OpenReview forum: "Learnable Tokenization for DNA Foundational Models"
_ICLR.cc/2026/Conference — ICLR 2026 Conference Withdrawn Submission_

### Official Review · Reviewer_nFMS · 2025-10-30

**Soundness:** 2
**Presentation:** 2
**Contribution:** 2
**Rating:** 2
**Confidence:** 4

**Summary:**

This paper adopts the adaptive tokenization idea from H-Net [1] and introduces two incremental improvements: (1) the pass-through of special tokens to the main model, and (2) a cross-attention-based dechunking scheme that replaces the original smoothing module. The authors name the resulting method DNACHUNKER and claim it achieves state-of-the-art performance in an old and easy benchmark. (In the following, I will refer to the two improvements as (1) and (2).

**Strengths:**

1.	The paper demonstrates a reasonable understanding of biological concepts. The interpretability analysis employs high phyloP scores and Repeat SINE, and Figure 3 uses a gene-track-style visualization that makes the results more accessible to biological researchers.

2.	Technically interesting but positioned as an incremental contribution. This paper proposes a learnable dynamic chunking approach. However, H-Net has already applied adaptive tokenization to DNA sequences, and prior work such as MxDNA [2] has also introduced the concept of adaptive DNA tokenization.

**Weaknesses:**

1.	From a machine learning perspective, the proposed improvements are incremental, and the evaluation is insufficient to demonstrate clear advantages. Adaptive and learnable tokenization methods in DNA modeling are not new—MxDNA, H-Net (which already includes DNA applications), and Life-Code [3] have each proposed distinct technical path. DNACHUNKER builds directly upon H-Net with two trick and yet provides limited ablation studies (only a brief one in Figure 2, compare DNACHUNKER and H-Net).

Improvement suggestions:

i. In Tables 1–2 and Figures 3–4, comparisons should include H-Net, w/o (1), and w/o (2) variants to quantify the contributions of the two claimed improvements.

ii. The authors should also compare with other adaptive tokenization methods—at minimum with MxDNA, a peer-reviewed and published approach. Similarly, biological interpretability comparisons in Figures 3–4 should include MxDNA for completeness.

2.	From the perspective of genomic model evaluation, the benchmarks used remain outdated, still focusing on short-sequence binary or ternary classification tasks typical of the DNABERT/DNABERT-2 era. Such simplistic feature-recognition tasks can often be solved by linear classifiers and no longer reflect current research standards.

By late 2025, evaluation of DNA foundation models has shifted toward regulatory genomics. For example:

The Nucleotide Transformer (published December 28, 2024) [4] introduced tasks such as eQTL, meQTL, and mutation effect prediction, showcasing the potential of large models for studying gene-regulatory mechanisms.

The July 24, 2025 paper “Evaluating the representational power of pre-trained DNA language models for regulatory genomics” [5] proposed a benchmark encompassing cell-type-specific functional genomics data for both DNA and RNA regulation.

This study revealed that current DNA pre-trained models often fail to produce embeddings that outperform raw k-mer representations.

Improvement suggestions:

i. Following [5], the authors could compare the pre-trained embeddings derived from k-mer, MxDNA, H-Net, and DNACHUNKER tokenizations when directly used as model inputs for downstream tasks, testing whether DNACHUNKER offers superior representational quality.

ii. Similarly, they could evaluate whether the output representations from models pre-trained with these different tokenizations (MxDNA, H-Net, DNACHUNKER) perform better than raw k-mer embeddings on downstream benchmarks [5].
Such evaluations would provide the genomics modeling community with clearer, knowledge-driven insights into the effectiveness and necessity of adaptive tokenization schemes.

Overall, the paper fails to provide new knowledge or meaningful contributions to either the machine learning or biological research. The method proposed in this paper is incremental, offering limited novelty or insight. The comparative experiments are incomplete, and the ablation study is too less to explain the advantages of the proposed improvements. The benchmark datasets used for evaluation are outdated and overly simplistic.

[1] Hwang S, Wang B, Gu A. Dynamic chunking for end-to-end hierarchical sequence modeling[J]. arXiv preprint arXiv:2507.07955, 2025.

[2] Qiao L, Ye P, Ren Y, et al. Model decides how to tokenize: Adaptive dna sequence tokenization with mxdna[J]. Advances in Neural Information Processing Systems, 2024, 37: 66080-66107.

[3] Liu Z, Li S, Chen Z, et al. Life-Code: Central Dogma Modeling with Multi-Omics Sequence Unification[J]. arXiv preprint arXiv:2502.07299, 2025.

[4] Dalla-Torre H, Gonzalez L, Mendoza-Revilla J, et al. Nucleotide Transformer: building and evaluating robust foundation models for human genomics[J]. Nature Methods, 2025, 22(2): 287-297.

[5] Tang Z, Somia N, Yu Y, et al. Evaluating the representational power of pre-trained DNA language models for regulatory genomics[J]. Genome Biology, 2025, 26(1): 203.

**Questions:**

Please refer to the points raised in the Weaknesses section.

---

### Official Review · Reviewer_qiYg · 2025-10-30

**Soundness:** 2
**Presentation:** 2
**Contribution:** 3
**Rating:** 6
**Confidence:** 3

**Summary:**

The paper introduces DNACHUNKER, an DNA language model that integrates a learnable dynamic DNA tokenization mechanism. The paper adapts the dynamic chunking mechanism for the masked language model with bi-directional Caduceus and cross-attention layers. The paper claims to achieve the state-of-the-art performance on DNA benchmarks.

**Strengths:**

1. The motivation is overall reasonable. Though the contribution of leveraging SSM is incremental, it is somewhat novel in DNA field.
2. The proposed method is logically solid and the experiment results that indicate the proposed contribution is the primary source of the performance gain.
3. The results demonstrate strong parameter efficiency and competitive accuracy across relevant benchmarks.
4. The manuscript is clearly written, and the architecture and pipeline figures effectively support and clarify the method.

**Weaknesses:**

1. The evaluation suite is not fully aligned with the claimed advantages, and it should add high-value tasks that directly probe robustness and multi-scale context such as SNV variant-effect prediction (ClinVar-style VEP), deep mutational scanning, MPRA promoter/enhancer assays, and long-range regulatory benchmarks.
2. The evidence base is incomplete because there is no head-to-head comparison against strong recent DNA LMs (e.g., Evo1, Evo2) with matched data, training tokens, and compute budgets.
3. Clarity and reproducibility can be improved by standardizing figure styles, adding explicit axis labels and units (bp), defining acronyms on first use.

**Questions:**

1. Can the authors add head-to-head comparisons against recent DNA LMs with matched data, training tokens, and compute budgets?
2. The authors claim emphasize robustness and multi-scale context. Can the authors include SNV variant-effect prediction and DMS assays to directly probe these properties?
3. How is the generalization of the proposed method across species or cell types (e.g., train on human, test on mouse/yeast) or across tasks without re-learning chunks?

---

### Official Review · Reviewer_54yt · 2025-10-31

**Soundness:** 2
**Presentation:** 2
**Contribution:** 2
**Rating:** 2
**Confidence:** 5

**Summary:**

This paper proposes DNACHUNKER, a learnable tokenization scheme for genomic sequences integrated into a bidirectional masked-language model. The model adapts token boundaries that adaptive chunks align better with functional heterogeneity.

**Strengths:**

1. Introduces a principled, learned tokenization for DNA with hierarchical chunking and de-chunking tightly coupled to a bidirectional MLM, beyond frequency-driven BPE and fixed k-mer schemes.
2. Good analysis for functional heterogeneity (smaller chunks in functional regions, larger ones in repetitive regions).

**Weaknesses:**

1. The paper argues that learned tokenization is more robust to small variation but does not include SNV-level benchmarks commonly used to test base-resolution modeling. Without comparisons to other DNA foundation models (e.g., NT, HyenaDNA, Evo), the central claim remains under-supported.

2. On Genomic benchmarks, important baselines are absent (e.g.,  NT and DNABERT-2). DNACHUNKER is not uniformly superior. The paper should show a complete leaderboard for the chosen tasks and discuss why it underperforms.

3. The method is motivated by efficiency and long-range dependency capture, yet there are no direct comparisons on long-context tasks (e.g., BEND or LRB benchmark).


4. No sensitivity to ratio-loss targets or boundary thresholding. The tokenization robustness metric uses hand-set mixing weights (e.g., for boundary Jaccard vs. content similarity) without sensitivity analysis.

**Questions:**

Please see Weaknesses.

---

### Official Review · Reviewer_ssKV · 2025-11-02

**Soundness:** 3
**Presentation:** 3
**Contribution:** 3
**Rating:** 4
**Confidence:** 4

**Summary:**

This work introduces DNAChunker, a learnable dynamic chunking mechanism that segments genomic sequences into variable-length fragments to enhance computational efficiency and robustness. The model employs a dual-layer encoder-decoder architecture combined with a cross-attention mechanism, achieving performance close to that of larger models in masked language modeling tasks.

Core Contribution: A learnable dynamic chunking mechanism is proposed, which adaptively segments sequences into variable-length fragments using a dual-layer encoder-decoder structure.

**Strengths:**

- Technically Complex and Comprehensive: The model features dynamic chunking, a hierarchical encoder-decoder structure, and a bidirectional Caduceus module.
- Strong Robustness: It exhibits strong robustness against sequence perturbations such as mutations, insertions, and shifts.
- High Computational Efficiency: Achieves comparable performance while significantly reducing the number of parameters, demonstrating high computational efficiency.

**Weaknesses:**

- Lack of Interpretability: The learned chunk boundaries are completely opaque, lacking validation of their association with biological functional regions.

- High Engineering Complexity: The training process is cumbersome, with significant costs associated with hyperparameter tuning. The paper lacks analysis of training stability and resource consumption.

- Weak Theoretical Motivation: There is no theoretical support for how dynamic chunking outperforms fixed k-mers in terms of information representation. The primary validation is conducted on masked language modeling (MLM) tasks, without demonstrating transferability or generalizability.

**Improvement Suggestions:**

- Conduct biological relevance validation for chunk boundaries (e.g., analysis of promoter and exon coverage).
- Supplement reports on training time, memory usage, and FLOPs to quantify efficiency gains.
- Demonstrate the model's transferability in generative or structural prediction tasks.
- Compare performance and interpretability differences across various chunk sizes.

If these issues can be addressed during the rebuttal period, I would consider raising my score. Besides, the titles in the manuscript (DNACHUNKER: LEARNABLE TOKENIZATION FOR DNA LANGUAGE MODELS) and in the system (Learnable Tokenization for DNA Foundational Models) are mismatched.

**Questions:**

please refer to the weaknesses part

---

### Note · Authors · 2025-11-15

**Comment:**

We sincerely thank the reviewers for the time and effort dedicated to evaluating our work. The feedback provided was insightful and constructive.

After careful consideration, we have concluded that the rebuttal period is insufficient to fully address the reviewers' concerns. Thus, we believe it is best to withdraw the submission at this time, and will incorporate the comments in our future manuscript. Thankyou once more for your feedbacks.

Sincerely, Authors.

**Withdrawal Confirmation:**

I have read and agree with the venue's withdrawal policy on behalf of myself and my co-authors.